# Parental Death and Premature Mortality in Individuals with Out-of-Home Care Experience in Sweden: A Nationwide Cohort Study

**DOI:** 10.3390/ijerph22040580

**Published:** 2025-04-07

**Authors:** Sandra Rogne, Ylva B. Almquist, Lars Brännström

**Affiliations:** 1Department of Social Work, Stockholm University, 10691 Stockholm, Sweden; lars.brannstrom@socarb.su.se; 2Department of Public Health Sciences, Stockholm University, 10691 Stockholm, Sweden; ylva.almquist@su.se

**Keywords:** out-of-home care, foster care, child welfare, childhood adversity, premature mortality, parental death, register data, cohort study, longitudinal, Sweden

## Abstract

Experiences of out-of-home care (placement in foster-family care or residential care) and parental death in childhood are known risk factors for premature all-cause mortality. However, it remains unclear whether parental death during placement moderates the association between out-of-home care and mortality, particularly when considering the timing and duration of placement. Longitudinal register data from 10 Swedish birth cohorts (*n* = 948,483) were analyzed. Around 2.5% (*n* = 23,628) had out-of-home care experience during ages 0–19. Sex-specific Cox proportional hazard regression models assessed associations between experience of out-of-home care (categorized by timing and duration), parental death, and premature all-cause mortality (ages 20–47). Both men and women with out-of-home care experience displayed increased risk of premature all-cause mortality, as did those who experienced parental death before age 20. However, statistical interaction analyses revealed no moderating effect of parental death on the association between placement and premature all-cause mortality. Compared to non-placed individuals, parental death during placement in out-of-home care did not further increase the risk of premature mortality across placement groups. Possible reasons include strong attachments within the out-of-home care setting or reduced stress towards biological parents. Further research is needed to explore the complex dynamics of parental loss within out-of-home care populations.

## 1. Introduction

Over the past few years, there has been increasing evidence suggesting that children removed from their families of origin by the child welfare services and placed in out-of-home care (OHC; including foster-family care and residential care) face higher risks of subsequent all-cause mortality, which refers to death from any cause [1]. Scholars have paid considerably less attention to adverse events occurring within the family of origin during the placement and the extent to which these may further influence the life chances of placed children. While the death of a biological parent is one such adverse event, it remains unclear what kind of influence it might have on the association between OHC and premature mortality, or preventable early death. Using large-scale, longitudinal register data from Sweden, this study aims to address this knowledge gap.

Despite the objective of providing children in OHC with a secure, safe, and more stable environment to facilitate healthier and more socially advantaged outcomes, the evidence presents a bleak picture of their life chances. Previous research has consistently shown that individuals with OHC experience face worse outcomes throughout their adult lives, including poverty, lower educational attainment, and weaker labor market attachment, as well as higher rates of criminality, homelessness, and health problems [2,3,4].

Of particular interest for the current study, they are also consistently reported to have higher risks of premature mortality [5,6]. For example, studies have revealed that individuals with OHC experience had twice the risk of all-cause mortality and were three times more likely to commit suicide in adulthood than their peers in the majority population [1]. A significantly elevated mortality risk has been noted even in comparison to individuals—including their own siblings—who had similarly adverse childhood experiences but were not placed in OHC [7,8,9]. Moreover, these findings are particularly pronounced among those with long-term OHC experience (often regarded as “growing up” in OHC) [8].

While the poorer life chances of individuals with OHC experience, including their increased mortality risks, are now well established, less attention has been given to the role of other potentially traumatic events occurring in the family of origin during the child’s placement. In this context, the death of a biological parent may be an event of particular significance [10,11]. Similar to those placed in OHC, children who experience parental death have been identified as a high-risk group, facing challenges such as school failure and low educational attainment, poverty and weak labor market attachment, health risk behaviors and substance misuse, criminality, psychiatric disorders, and premature mortality [12,13,14,15].

Past research has indeed demonstrated a strong correlation between OHC and parental death. For example, an earlier study found that a significant number of individuals with OHC experience entered adulthood without their biological parents still alive: over 25% of them had lost at least one parent before the age of 18, compared to 3–4% in the general population [16]. The overlap between the populations of children experiencing OHC and parental death is not unexpected, considering that the death of a biological parent in itself constitutes a risk factor for being placed in OHC [17]. Additionally, studies indicate that the separation of a child from their parent can cause severe emotional distress in parents, particularly for mothers, leading to a deterioration in mental health that, in turn, increases risk of premature mortality for the parent [18].

To some extent, the correlation between OHC and parental death may also be attributed to the same underlying causes, such as parental substance misuse or psychiatric problems, often coupled with socioeconomic disadvantages [16]. For instance, the risk of substance misuse-related deaths among biological mothers and fathers has been found to increase with the duration of OHC placement for their children [16]. It has been suggested that most children in long-term placement have at least one biological parent who misuses alcohol or illicit drugs. Consequently, deaths related to parental substance misuse are more common, due to the elevated risks not only from substance misuse itself but also from suicide, accidents, or violence [16].

The fact that there is an overlap between the populations of children experiencing OHC and parental death, sharing many common causes and consequences, does not allow for any conclusions about whether parental death might moderate the association between OHC and premature mortality. On the one hand, in line with existing theories on the cumulative effects of childhood adversity [19], it may be reasonable to expect that children who experience the death of a biological parent while in OHC could face even more elevated mortality risks compared to those placed children who do not encounter such a loss. Children in OHC are already a highly vulnerable population, and the additional burden of a parent’s death might increase their rates of mental and behavioral problems, which could in turn affect their mortality risks. Moreover, the death of a biological parent most likely decreases the chances of family reunification, thereby prolonging the child’s placement. This prolonged placement may negatively affect life chances, considering that long-term placements are linked particularly strongly to premature mortality.

On the other hand, given the complexity of child–parent relationships in OHC situations and the often-high levels of adversity that prompted child welfare services to intervene initially, parental death may not significantly add to the child’s burden. It is also worth noting that in Sweden, where this study is set, the child welfare system emphasizes a family service orientation, with reunification with biological parents being a primary objective for children in care [20]. Parental visitation is considered crucial for maintaining the child–parent relationship [21]. Nevertheless, continued contact with biological parents can introduce emotional stress, and conflicting loyalty issues may arise as the attachment to the foster family grows stronger while the child tries to preserve connections with biological parents [21,22,23]. Accordingly, it is possible that parental death even has a weaker association with premature mortality in OHC populations than among individuals who were not placed in care.

Using linked longitudinal register data from 10 Swedish birth cohorts (n > 1,000,000), of which approximately 2.5% have experienced OHC at some point during their childhood, this study aims to assess the possible contribution of parental death to the relationship between OHC and premature all-cause mortality. To set the stage for this examination, the first step is to empirically analyze the associations between OHC and premature mortality on the one hand, and parental death and premature mortality on the other. Building on this empirical framework, the study investigates two opposing hypotheses: (a) children in OHC who experience parental death have higher risks of mortality compared to children in OHC who do not experience parental death, and (b) parental death does not add to the mortality risks to any greater extent among OHC-experienced individuals compared to those who were not placed in care. Since no prior study has addressed these opposing hypotheses, this research fills an important gap in the literature on mortality in OHC populations. Given that premature mortality is more common in men compared to women [24], and to allow for sex-specific responses to OHC experience as well as parental death, results from sex-stratified analyses are reported. Additionally, since mortality risks within the OHC population have been found to vary according to duration and timing of placement, several subgroups are identified and examined.

## 2. Materials and Methods

### 2.1. Study Design

This study has a prospective design based on linked longitudinal register data. For the purposes of this study, data from 10 birth cohorts spanning from 1972 to 1981, who were alive and residing in Sweden in 1985 according to the Population and Housing Census, were used. The cohorts were tracked in the registers from birth until the date of death or, at most, until 31 December 2018 (ages 37–47). The linkage between the cohort members and their biological parents was facilitated by the Multigenerational Register (held by Statistics Sweden). Ethical permission for the study was obtained from the Swedish Ethical Review Authority (no. 2020–00250). All accessed data are pseudonymized (i.e., person non-identifiable). This means that the authors never had access to information that could identify individual participants during or after data extraction.

### 2.2. Study Population

The study population initially consisted of 1,028,957 individuals (527,034 men and 501,923 women). After accounting for missing data (*n* = 80,474/7.8% of the population), primarily due to incomplete information on key variables such as parental socioeconomic factors, the final analytical sample included 948,483 individuals (486,841 men and 461,642 women), of which approximately 2.5% (*n* = 23,628) had some experience of OHC between the ages of 0 and 19 (15,137 men and 13,994 women).

### 2.3. Variables

The outcome variable in all analyses was premature all-cause mortality among the cohort members. The follow-up period for mortality began when the individual reached the age of 20 and continued until they reached age 47, specifically between 1 January 1992 and 31 December 2018. Individuals who died before the age of 20 years were excluded from the analysis (*n* = 2416/0.24% of the population). Information on mortality was obtained from the Swedish Cause of Death Register (held by the National Board of Health and Welfare).

Records on OHC were retrieved from the Swedish Child Welfare Intervention Register (held by the National Board of Health and Welfare). Individuals with experience of OHC were classified into mutually exclusive groups representing various combinations of timing and duration of placement [25]. This approach has been applied in numerous prior Swedish studies and has demonstrated strong discriminatory capability across a range of outcomes [26,27,28,29]. The categorization of placements was used to divide cohort members into the following groups:No OHC: No records of OHC placement;Early short-term placement: First placement before the age of 13 for less than one year;Early intermediate placement: First placement before the age of 13 for one to five years;Long-term placement: First placement before the age of 13 for more than five years;Teenage placement: First placement after the age of 13, without any specification of time in OHC.

These OHC groups differ from each other in important ways. Individuals placed in OHC as teenagers are significantly different, as many are placed due to their own behavior rather than parental behavior [30,31]. Those who enter OHC before the age of 13 are typically placed due to parental abuse or neglect, while teenage placement is often a consequence of juvenile delinquency and/or substance abuse [32]. Individuals with long-term experience in OHC also differ from the others, as they have been placed in care for a substantial period. These children essentially “grow up” in OHC until they eventually age out of care upon reaching the age of majority or completing upper secondary education [20,29].

Information about parental death (ages 0–19) was retrieved from the Cause of Death Register. A decision was made to combine the death of one or both parents into a single category, as only 0.1% of the analytical sample had experienced the death of both parents before the age of 20. The reason for limiting parental death to occurrences when the child was aged 0–19 years is because this period aligns with when a child could be placed in OHC.

Guided by prior studies on social selection into OHC [33], the analysis encompassed various confounding factors concerning both the cohort members and their biological parents, identified in prior literature. Individual confounders referred to birth year (retrieved from the Medical Birth Register held by the National Board of Health and Welfare) and country of birth (retrieved from the Total Population Register held by Statistics Sweden). To account for the diverse backgrounds of the study population, country of birth was categorized into four mutually exclusive groups: “Sweden”, “other Nordic country”, “Europe (excluding the Nordic countries)”, and “outside Europe”.

Demographic and socioeconomic circumstances (retrieved from the Longitudinal Integrated Database for Health and Labor Market Studies held by Statistics Sweden) for the mother and the father were also included. Maternal and paternal confounders included country of birth (retrieved from the Total Population Register held by Statistics Sweden), which included the same four categories as for the individual. Further confounders included educational attainment, poverty, single parenthood, criminality, substance abuse, and mental health problems.

Level of education was originally coded into a seven-grade scale according to the Swedish educational classification SUN 2000, with a range from compulsory education to postgraduate education. For this study, a new variable was made with two categories: low and high. Lower educational level (including a “no information” category) referred to compulsory education as its highest level (nine years or lower) while higher education included all higher levels. Poverty referred to means-tested social assistance, and it was coded as yes in the instance of at least one record of receiving social welfare. Single parenthood was coded as yes or no.

Confounders linked to psychopathological characteristics of the mother and the father were also included, facilitated by data from the Patient Register (held by the National Board of Health and Welfare) and the Conviction Register (held by the National Council for Crime Prevention). Criminality was defined by at least one conviction that was followed by a sentence to probation, prison, or psychiatric care. Substance abuse was coded as “yes” if there had been registered at least one hospitalization due to alcohol and/or narcotics-attributable causes according to standardized International Classification of Diseases/ICD, either primary or secondary diagnosis (ICD10: F101-F109, F11, F12, F14, F16, F19, G621, I426, K70, K292, O355, P044, T436, Z503, Z715, Z722; ICD 9: 291, 292, 303, 304, 305A, 357F, 425F, 535D, 571A, 571B, 571D, 965A, 968F, 969G, 969H). Mental health problems were coded as “yes” if there was a record of at least one hospitalization with a psychiatric primary or secondary diagnosis according to standardized ICD codes (ICD 10: F00-F99, apart from F10-F16, F18 and F19; ICD 9: 290–319, apart from 291, 292, 303, 304 and 305A).

Due to data limitations, the confounding factors associated with maternal and paternal socioeconomic conditions mostly pertain to circumstances in 1990, when the cohort members were aged between 11 and 18 years old. This is potentially problematic, particularly for the OHC groups where the individuals entered care before the age of 13. Yet, these confounders should offer a satisfactory window into maternal and paternal circumstances even for the pre-care timeframe. For instance, in a Swedish context, parenthood tends to be negatively associated with continued educational attainment [34]. Educational level can therefore be considered a fairly stable measurement.

Regarding measurements of criminality, substance abuse, and mental health problems, the whole observational period was utilized (1972–2018). Hospitalization due to both substance abuse and mental health problems implies a certain severity level, which can indicate long-term problems even prior to hospital care. This, in turn, indicates problematic conditions in the birth home environments of the cohort members. The same argument can be made regarding criminality in the context of the Swedish court system, as a series of smaller offences is usually the backdrop for a sentence of probation or prison [35]. While not ideal, employing such an extended measurement period is a commonly adopted practice in Scandinavian register-based studies [36,37,38].

### 2.4. Statistical Analysis

Standard descriptive statistics of the study variables for men and women are reported separately for OHC experienced and non-OHC experienced (percentage). Average follow-up (mean, median, min–max) for the study participants across the OHC and non-OHC groups can be found in Appendix A. Cox proportional hazard regression [39] was employed to analyze the association between OHC and premature all-cause mortality, as well as between parental death and premature all-cause mortality. Subjects were entered on the day of their birth. As noted above, the observational time for each subject began at age 20 and continued until either the date of death or the end of the follow-up period on 31 December 2018. Hazard ratios (HRs) with 95% confidence intervals (CIs) were generated for two sets of models: the first set was unadjusted, while the second set was adjusted for all confounding factors.

In addition, statistical interaction analysis (effect measure modification) was performed to examine whether parental death moderated the association between OHC and premature all-cause mortality, i.e., whether the association between one exposure and outcome varied within strata of the other exposure [40]. Here, the main effect terms (OHC group and parental death) as well as the interaction term (OHC group x parental death) were included, after which predictive margins were produced. This approach was supplemented with a comparison of model fit. If a likelihood ratio (LR) test produced a *p*-value below 0.05, the model with the interaction term was deemed to fit the data better than a model without the interaction term.

Separate Cox regression models were conducted for men and women, based on sex assigned at birth. The proportional hazards assumption was evaluated using Schoenfeld residuals, and there was no evidence of a violation against the proportionality assumption in any of the models. To address issues with tied failure times, various methods were employed, including the Breslow approximation, Efron approximation, exact marginal calculation, and exact partial calculation. All these methods yielded consistent results, indicating no problems associated with tied failure times. Data management and statistical analysis were performed using Stata 17/SE version, and the margins and marginsplot commands were utilized to calculate predictive margins and visualize the results from the statistical interaction analysis [41].

## 3. Results

Descriptive statistics reported in Table 1 show that premature mortality was more frequent in men than in women in both general and OHC groups, with larger differences in the OHC population. Parental death before age 20 was observed in about 2% of both sexes in the general population and in around 5% in the OHC population. “Teenage placement” was the primary OHC type, followed by “early short-term”, “long-term”, and “early intermediate” placements for both sexes. OHC individuals had consistently higher rates of maternal and paternal confounding factors relating to foreign background, single parenthood, lower education, poverty, substance abuse, mental health issues, and criminality. For instance, the prevalence of maternal poverty in OHC men and women was around 40 percentage units higher compared to their non-OHC counterparts. Similarly, the difference in paternal substance abuse was around 23 percentage units.

The unadjusted regression models presented in Table 2 show that both men and women placed in OHC had a higher risk of premature mortality compared to their general population non-OHC peers, regardless of OHC group. In the female OHC population, the risk increased in relation to the timing and duration of placement. The risk ranged from more than two times higher (HR = 2.36, 95% CI = 1.78; 3.13) for the early short-term placement group (first placement before the age of 13 and for up to one year) to nearly five times higher (HR = 4.70, 95% CI = 4.00; 5.49) for individuals placed in OHC as teenagers.

For the male population, the lowest and highest risk levels were similar to those of the female population. However, it is noteworthy that men placed in OHC as teenagers had over seven times higher risk (HR = 7.49, 95% CI = 6.86; 8.19) of premature mortality compared to their general population peers, which was significantly higher than any of the other OHC groups. Furthermore, both men (HR = 1.70, 95% CI = 1.49; 1.92) and women (HR = 1.49, 95% CI = 1.29;1.71) had a higher risk of premature mortality if they had experienced the death of a parent before the age of 20, compared to their general population peers without this experience. After including the confounding factors in the adjusted model, all estimates were reduced, but the main conclusions remained the same.

Table 3 reports the results from models including main effects as well as statistical interaction effects. Predictive margins (in this case, HRs) based on the statistical interaction analyses depicted in Table 3 are presented in Appendix A. Similar to the results presented in Table 3, the models with unadjusted predictive margins revealed a higher risk of premature all-cause mortality in every OHC group for both men and women, regardless of the experience of parental death, compared to individuals without experiences of OHC and parental death.

Furthermore, as indicated by the wide 95% CIs for the interaction terms across OHC groups, there were no significant differences in the risk of premature all-cause mortality between individuals who had and had not experienced parental death. Although adjusting for confounding factors resulted in a notable reduction in the estimates, the main conclusions remained unchanged. These findings are visually depicted in Figure 1 (panel A: men; panel B: women), clearly demonstrating the absence of any moderation of the association between OHC and premature all-cause mortality by parental death. Moreover, the adjusted models reported in Table 3, which included an interaction term, did not improve model fit compared to the adjusted models without an interaction term (LR test men: *p* = 0.221; LR test women: *p* = 0.219). This further indicates that there is no evidence that parental death moderates the association between OHC experience and premature all-cause mortality.

## 4. Discussion

Using longitudinal register data for nearly 1,000,000 Swedes born between 1972 and 1981, of whom around 2.5% had childhood experience of OHC, this study aimed to enhance our understanding of the association between parental death and premature all-cause mortality in OHC populations. More specifically, it tested two opposing hypotheses regarding the link between OHC experience and parental death during placement. After adjusting for confounding factors, the results consistently showed a higher risk of premature all-cause mortality in both men and women with OHC experience. This elevated risk was observed regardless of the timing and duration of placement, aligning with previous research [5,7].

Specifically, the risk was particularly pronounced for individuals placed in OHC during their teenage years. Although the reasons for and types of placement were not determined in this study, residential care has been more prevalent among the OHC population placed in care as adolescents, which has been known to have more negative outcomes compared to placement in foster family care [42]. Individuals in teenage placement are considered to be a high-risk group in many respects, often placed due to their own detrimental behaviors, such as drug abuse or violence, rather than parental adverse behaviors [30,31]. The elevated risk of premature mortality in the teenage placement category is consistent with previous studies, which also show lower levels of educational attainment and income [30,43].

While a higher risk of premature all-cause mortality was found among those who have experienced parental death, a novel finding of the current study was that parental death during placement in OHC did not increase the mortality risk. Although parental death can still be traumatic for children in OHC, there are possible explanations for why it seems to have a lesser influence on premature all-cause mortality.

Children in OHC do not experience firsthand the same issues faced by the general population living with their parents at the time of parental death. After the death of a parent, the surviving parent may face mental health challenges that impact their caregiving abilities [44]. There is also the risk of financial strain on the family following the death of one or both parents, particularly with the death of a father, leading to socioeconomic disadvantages that create another long-term risk factor for the child [44,45]. As children in OHC live with another family or in residential care, they may be shielded from these risks, potentially making OHC a protective factor in some cases. However, these explanations are speculative and require further investigation.

Furthermore, one can speculate that reduced stress towards biological parents and the importance of quality attachment with the placement family or a reliable adult in residential care can play a significant role in how a child copes with trauma and adversities, thus moderating the impact of parental loss. Attachment theory suggests that early experiences with biological parents shape one’s expectations and beliefs about the availability and responsiveness of others in times of distress [46]. Insecure attachment patterns, characterized by anxiety or avoidance, may lead to greater vulnerability to stress and poorer health outcomes [47].

However, the quality of the placement family or residential group can serve as a protective factor for those who have experienced disrupted attachment relationships prior to OHC [48]. Positive relationships with caring adults predict better outcomes after OHC, providing a secure and supportive attachment [49], which fosters a sense of security and trust in others, mitigating the impact of stressors such as parental death [48]. The quality of the placement and the reduced emotional stress towards biological parents may contribute to a more stable and predictable environment, alleviating feelings of anxiety and the conflicting feelings of loyalty that are known to arise in children in OHC [47].

This population-based study has several strengths, including a large sample size that enables the investigation of smaller population segments and rare events, yielding nationally representative results. The use of register-based mortality follow-up minimized attrition, enhancing the reliability of the findings. The sample encompasses individuals with diverse timing and durations of OHC placements.

However, the specific event of parental death during OHC placement was relatively rare, particularly in subgroup analyses. Despite this, it remained important to examine and report results by placement group due to significant differences in their characteristics. Both OHC experience and parental death independently predicted a higher risk of premature mortality compared to the general population, aligning with existing evidence. The use of prospectively gathered data from long-standing care records improves the validity compared to self-reported data, especially for sensitive issues like childhood separation from biological parents. Furthermore, the nature of the sample allowed for the adjustment of a wide range of confounding factors.

Like any scientific study, this research has certain limitations. Register-based data may not fully capture the complex processes underlying the observed associations. Information on factors such as personality traits or the overall quality of OHC is often not available in register-based datasets. Despite controlling for several confounding factors, there may still be unmeasured variables that influenced the results. The study did not account for specific causes of OHC placement, type of placement (whether foster-family care or residential care), genetic risk factors, social networks, the child’s relationships with biological parents, foster parents, or other children in residential care, nor whether there was continued contact with the biological family during OHC prior to the parental death.

The findings of this study may not be generalizable to individuals born in more recent years, as the cohorts included were born between 1972 and 1981. However, there were no known changes to the OHC system or register data collection methods during this period that would impact the study’s reliability. It should be noted that this study is based on Swedish registers, which may limit the applicability of the findings to contexts outside of Nordic and Western countries. Unlike several other countries, placement during teenage years is more common than at younger ages in Nordic countries [43,50], a pattern that was also observed in this study.

Another limitation concerns the subgroup of children in the OHC population who had already experienced parental death before entering OHC. In the sample, 2386 individuals had experienced the death of one (*n* = 2225) or both (*n* = 161) biological parents prior to placement. Previous evidence suggests an overrepresentation of bereaved children in OHC, and that parental death may have influenced the decision to place these children in care [16]. Due to data limitations, this could not be confirmed in the current study. However, this subgroup was included in the analysis. This inclusion is justified because the individuals who had lost one parent prior to placement could also have lost a parent during placement, making them still at risk and a relevant part of the sample. The remaining 161 individuals who were bereaved of both parents before placement represent a small but potentially significant limitation in the results.

Nevertheless, the frequency of premature mortality within the OHC group that experienced parental death prior to placement (5.71%) was similar to that in the group that was bereaved of one parent during placement (5.52%). The individuals who were bereaved of both parents before placement represented only six cases and a slightly lower percentage of premature mortality (3.73%) compared to those who experienced it during placement (4.79%). Furthermore, we acknowledge that the large imbalance in sample sizes between the non-OHC and OHC groups may have contributed to the lack of statistically significant findings in some analyses. While this is an inherent issue in studies of out-of-home care due to the relatively small proportion of the population with such experiences, it is important to consider this when interpreting the results.

This study provides insights into the associations between OHC, parental death, and premature mortality. While individuals with a history of OHC face elevated risks for premature mortality and are disproportionately likely to experience parental death, the findings indicate that parental death does not substantially nor uniformly alter the risk of premature mortality in OHC populations. However, further research is necessary to understand the complexities of OHC placement and parental death more fully, including the roles of biological parental involvement, relationships and attachments, loyalty conflicts, and placement duration.

Future studies should consider the reasons for placement, type of placement, age at parental death, cause of death, gender differences, and the dynamics between the child and the foster family. Such research will deepen our understanding of how parental death impacts the OHC population, thereby benefiting child welfare and public health efforts.

The findings of this study carry significant implications for child welfare policy and practice, as well as for public health concerns regarding the long-term impact of adverse childhood experiences on premature mortality. The heightened risk of premature all-cause mortality among the OHC population, regardless of the timing and duration of placement, underscores the urgent need to improve the quality of OHC for individuals across all ages and placement groups. Policymakers should prioritize initiatives that enhance placement stability and provide resources to support the transition to independent adulthood for children and youth in OHC [51]. Emphasizing the quality of foster-family homes and residential care facilities can facilitate positive relationships with caregivers, aiding individuals in coping with adverse experiences and leading to better long-term outcomes for the OHC population.

To support parentally bereaved children in OHC and mitigate the long-lasting risk factors associated with parental death [12], targeted health and social support should be provided. This support is crucial regardless of the moderating effect of parental death on the association between OHC and premature mortality. Given that society assumes parental responsibilities for these children, addressing the specific challenges related to childhood parental death is essential.

Additionally, there is a pressing need for after-care services in Sweden to support the transition from OHC to independent adulthood, especially for youth who have experienced the loss of one or both biological parents before leaving care [3]. Currently, Sweden lacks such services, leaving these individuals with limited resources and lacking emotional and practical support during this critical transition period [52]. By combining high-quality OHC that fosters supportive relationships and offering resources for a smoother transition, there is potential to significantly improve long-term outcomes for the OHC population.

## 5. Conclusions

This study demonstrates that individuals with OHC experience face an increased risk of premature all-cause mortality compared to those without OHC experience. Similarly, parental death before age 20 is associated with heightened all-cause mortality risks. However, our findings indicate that parental death during OHC placement does not further increase mortality risks beyond those associated with OHC alone. Potential explanations include protective factors within OHC settings or altered attachments to biological parents. These results highlight the complexity of the OHC experience and emphasize the need for further research, as well as the development of policies and practices within the child welfare system, where society assumes parental responsibilities and ensures the safety and well-being of children. Further, additional research involving nationwide samples that include a substantial OHC population from other countries is essential to establish the external validity of the conclusions presented in this study.

## Figures and Tables

**Figure 1 ijerph-22-00580-f001:**
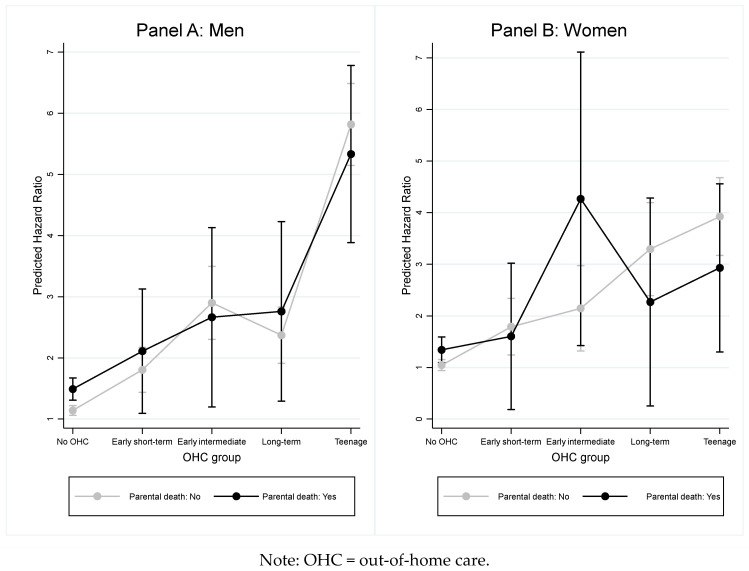
Adjusted predictive margins with 95% confidence intervals.

**Table 1 ijerph-22-00580-t001:** Sample characteristics, by OHC experience and sex.

	Men (*n* = 486,841)				Women (*n* = 461,642)			
	OHC		No OHC		OHC		No OHC	
Variables	*n*	%	*n*	%	*n*	%	*n*	%
Cohort members								
Premature all-cause mortality								
No	11,339	92.2	465,925	98.1	10,969	96.9	446,263	99.1
Yes	964	7.8	8613	1.8	356	3.1	4054	0.9
Parental death								
No	11,660	94.8	465,252	98	10,728	94.7	441,603	98
Yes	643	5.2	9286	2	597	5.3	8714	2
Out-of-home care (OHC)								
No placement			474,538	97.5			450,317	97.5
Early short-term placement	3015	0.6			2789	0.6		
Early intermediate placement	1760	0.4			1381	0.3		
Long-term placement	2436	0.5			2070	0.5		
Teenage placement	5092	1.1			5085	1.1		
Birth year								
1972	1603	11.9	51,458	10.8	1242	11	48,786	10.8
1973	1456	10.8	50,979	10.7	1151	10.1	48,378	10.7
1974	1402	10.4	51,647	10.9	1159	10.2	48,943	10.9
1975	1318	9.8	48,646	10.3	1120	9.9	46,464	10.3
1976	1245	9.2	46,372	9.8	1028	9.1	43,808	9.7
1977	1203	9	45,544	9.6	986	8.7	42,778	9.5
1978	1244	9.2	43,859	9.2	1058	9.3	41,767	9.2
1979	1336	10	45,462	9.6	1168	10.3	43,224	9.6
1980	1372	10.1	45,988	9.7	1218	10.8	43,592	9.6
1981	1317	9.8	44,583	9.4	1195	10.6	42,577	9.4
Birth country								
Sweden	11,865	96.4	466,191	98.2	11,011	97.2	442,432	98.3
Other Nordic country	112	0.9	1921	0.4	84	0.7	1803	0.4
Europe (excl. the Nordic countries)	110	0.9	2937	0.6	71	0.6	2687	0.6
Outside Europe	216	1.8	3489	0.7	159	1.4	3395	0.8
Mothers								
Educational level								
Low	5686	46.2	126,457	26.7	5291	46.7	120,326	26.7
High	6617	53.8	348,081	73.3	6036	53.3	329,991	73.3
Poverty								
No	6756	54.9	449,110	94.6	6180	54.6	426,287	94.7
Yes	5547	45	25,428	5.4	5145	45.4	24,030	5.3
Single parenthood								
No	7129	58	406,848	85.7	6812	60.2	383,678	85.2
Yes	5174	42	67,690	14.3	4513	39.8	66,639	14.8
Criminality								
No	12,115	98.5	474,123	99.9	11,160	98.5	449,918	99.9
Yes	188	1.5	415	0.09	156	1.5	399	0.09
Substance abuse								
No	9454	76.8	459,531	96.8	8586	75.3	436,220	96.9
Yes	2849	23.2	15,007	3.2	2799	24.7	14,097	3.1
Mental health problems								
No	8450	68.7	443,326	93.4	7521	66.4	421,043	93.5
Yes	3853	31.3	31,212	6.6	3804	33.6	29,274	6.5
Birth country								
Sweden	9897	80.4	424,784	89.5	9251	81.7	403,373	89.6
Other Nordic country	1253	10.2	25,228	5.3	1190	10.5	23,758	5.3
Europe (excl. the Nordic countries)	755	6.1	17,498	3.7	569	5	16,510	3.7
Outside Europe	398	3.2	7028	1.5	315	2.8	6676	1.5
Fathers								
Educational level								
Low	5691	46.3	152,724	32.2	5220	46.1	145,150	32.2
High	6612	53.7	321,814	67.8	6105	53.9	305,167	67.8
Poverty								
No	8522	69.3	452,581	95.4	7974	70.4	429,530	95.4
Yes	3781	30.7	21,957	4.6	3351	29.6	20,787	4.6
Single parenthood								
No	10,820	88	453,408	96	10,013	88.4	432,532	96.1
Yes	1483	12	21,130	4	1312	11.6	17,785	3.9
Criminality								
No	11,838	96.2	471,154	99.2	10,857	95.9	447,006	99.3
Yes	465	3.8	3384	0.8	468	4.1	3311	0.7
Substance abuse								
No	8526	69.3	437,942	92.3	7962	70.3	415,007	92.2
Yes	3777	30.7	36,596	7.7	3363	29.7	35,310	7.9
Mental health problems								
No	10,102	82.1	447,966	94.4	9334	82.4	424,718	94.3
Yes	2201	17.9	26,572	5.6	1991	17.6	25,599	5.7
Birth country								
Sweden	9576	77.9	423,629	89.3	9047	79.9	401,824	89.2
Other Nordic country	1137	9.2	20,055	4.2	1025	9	19,091	4.2
Europe (excl. the Nordic countries)	966	7.9	21,960	4.6	780	6.9	20,998	4.7
Outside Europe	624	5.1	8894	2	473	4.2	8404	1.9

Note: OHC = out-of-home care.

**Table 2 ijerph-22-00580-t002:** Associations between OHC, parental death, and premature mortality. Results from Cox regression analysis, by sex.

	Men (*n* = 486,841)			Women (*n* = 461,642)		
		HR (95% CI)	HR (95% CI)		HR (95% CI)	HR (95% CI)
Variables	Deaths (*n*)	Unadjusted	Adjusted	Deaths (*n*)	Unadjusted	Adjusted
OHC						
No OHC (Ref)	9071	1.00	1.00	4278	1.00	1.00
Early short-term	149	2.50 (2.08–3.01)	1.48 (1.23–1.79)	73	2.36 (1.78–3.13)	1.66 (1.24–2.22)
Early intermediate	153	4.30 (3.56–5.20)	2.24 (1.84–2.72)	56	3.51 (2.50–4.92)	2.21 (1.561–3.14)
Long-term	217	3.63 (3.06–4.32)	1.74 (1.45–2.09)	110	4.45 (3.48–5.69)	2.74 (2.10–3.57)
Teenage	636	7.49 (6.86–8.19)	4.61 (4.19–5.07)	198	4.70 (4.00–5.49)	3.57 (3.02–4.23)
Parental death						
No (Ref)	9450	1.00	1.00	4339	1.00	1.00
Yes	720	1.70 (1.49–1.92)	1.29 (1.14–1.47)	349	1.49 (1.29–1.71)	1.26 (1.10–1.45)
Individual confounders						
Birth year						
1972 (Ref.)	-	**-**	1.00	-	**-**	1.00
1973	-	**-**	1.03 (0.95–1.13)	-	**-**	0.99 (0.87–1.12)
1974	-	**-**	1.05 (0.96–1.15)	-	**-**	1.05 (0.92–1.19)
1975	-	**-**	0.98 (0.90–1.08)	-	**-**	1.06 (0.93–1.22)
1976	-	**-**	1.05 (0.95–1.16)	-	**-**	1.00 (0.87–1.15)
1977	-	**-**	1.17 (1.06–1.29)	-	-	1.03 (0.89–1.19)
1978	-	**-**	1.15 (1.04–1.27)	-	-	0.88 (0.75–1.03)
1979	-	**-**	1.19 (1.08–1.31)	-	-	0.83 (0.71–0.97)
1980	-	**-**	1.22 (1.10–1.35)	-	-	1.01 (0.87–1.18)
1981	-	**-**	1.20 (1.08–1.33)	-	-	1.05 (0.89–1.23)
Birth country						
Sweden (Ref.)	-	**-**	1.00	-	-	1.00
Other Nordic country	-	**-**	1.06 (0.82–1.38)	-	-	1.01 (0.66–1.55)
Europe	-	**-**	0.74 (0.55–1.01)	-	-	0.99 (0.63–1.56)
Outside Europe	-	**-**	0.20 (0.08–1.33)	-	-	1.21 (0.74–2.00)
Maternal confounders						
Educational level						
Low (Ref.)	-	**-**	1.00	-	-	1.00
High	-	**-**	0.87 (0.82–0.91)	-	-	0.94 (0.87–1.02)
Poverty						
No (Ref.)	-	**-**	1.00	-	-	1.00
Yes	-	**-**	1.34 (1.23–1.45)	-	-	1.13 (0.99–1.29)
Single parenthood						
No (Ref.)	-	**-**	1.00	-	-	1.00
Yes	-	**-**	1.24 (1.17–1.31)	-	-	1.19 (1.10–1.30)
Criminality						
No (Ref.)	-	**-**	1.00	-	-	1.00
Yes	-	**-**	1.10 (0.76–1.61)	-	-	0.90 (0.45–1.82)
Substance abuse						
No (Ref.)	-	**-**	1.00	-	-	1.00
Yes	-	**-**	1.21 (1.11–1.33)	-	-	1.12 (0.97–1.30)
Mental health problems						
No (Ref.)	-	**-**	1.00	-	-	1.00
Yes	-	**-**	1.37 (1.27–1.47)	-	-	1.42 (1.27–1.58)
Birth country						
Sweden (Ref.)	-	**-**	1.00	-	-	1.00
Other Nordic country	-	**-**	1.18 (1.08–1.30)	-	-	1.26 (1.09–1.45)
Europe	-	**-**	1.03 (0.90–1.18)	-	-	0.95 (0.77–1.19)
Outside Europe	-	**-**	0.80 (0.62–1.03)	-	-	0.72 (0.47–1.11)
Paternal confounders						
Educational level						
Low (Ref.)	-	**-**	1.00	-	-	1.00
High	-	**-**	0.89 (0.85–0.94)	-	-	0.92 (0.86–0.99)
Poverty						
No (Ref.)	-	**-**	1.00	-	-	1.00
Yes	-	**-**	1.14 (1.04–1.24)	-	-	1.05 (0.91–1.21)
Single parenthood						
No (Ref)	-	**-**	1.00	-	-	1.00
Yes	-	**-**	1.16 (1.06–1.26)	-	-	1.02 (0.88–1.18)
Criminality						
No (Ref.)	-	**-**	1.00	-	-	1.00
Yes	-	**-**	1.24 (1.04–1.47)	-	-	1.08 (0.81–1.46)
Substance abuse						
No (Ref.)	-	**-**	1.00	-	-	1.00
Yes	-	**-**	1.37 (1.28–1.47)	-	-	1.21 (1.09–1.35)
Mental health problems						
No (Ref.)	-	**-**	1.00	-	-	1.00
Yes	-	**-**	1.24 (1.15–1.34)	-	-	1.17 (1.03–1.32)
Birth country						
Sweden (Ref.)	-	**-**	1.00	-	-	1.00
Other Nordic country	-	**-**	1.29 (1.17–1.43)	-	-	1.10 (0.93–1.29)
Europe	-	**-**	1.09 (0.96–1.23)	-	-	1.02 (0.84–1.22)
Outside Europe	-	**-**	1.38 (1.16–1.65)	-	-	1.09 (0.81–1.48)

Note: OHC = out-of-home care; Ref. = reference category; HR = hazard ratio; CI = confidence interval.

**Table 3 ijerph-22-00580-t003:** OHC and premature mortality: moderation by parental death. Main effects and statistical interaction effects. Results from Cox regression analysis, by sex.

	Men (*n* = 486,841)		Women (*n* = 461,642)	
		HR (95% CI)		HR (95% CI)
Variables	Deaths (*n*)	Adjusted	Deaths (*n*)	Adjusted
OHC				
No OHC (Ref.)	9071	1.00	4278	1.00
Early short-term	149	1.47 (1.21–1.79)	73	1.69 (1.26–2.27)
Early intermediate	153	2.28 (1.87–2.79)	56	2.08 (1.43–3.03)
Long-term	217	1.80 (1.48–2.14)	110	2.84 (2.17–3.71)
Teenage	636	4.69 (4.26–5.17)	198	3.68 (3.10–4.36)
Parental death				
No (Ref.)	9450	1.00	4339	1.00
Yes	720	1.35 (1.18–1.55)	349	1.42 (1.14–1.75)
Interaction OHC and parental death				
No OHC + no parental death (Ref.)	8492	1.00	3983	1.00
Early short-term + parental death	19	0.99 (0.49–1.97)	10	0.64 (0.15–2.68)
Early intermediate + parental death	21	0.66 (0.28–1.51)	11	1.45 (0.55–3.84)
Long-term + parental death	25	0.60 (0.24–1.48)	13	0.43 (0.10–1.78)
Teenage + parental death	76	0.67 (0.44–1.02)	20	0.47 (0.19–1.18)

Note: OHC = out-of-home care; Ref. = reference category; HR = hazard ratio; CI = confidence interval. Control estimates become the same as in Table 2.

## Data Availability

The dataset for this study is based on linked administrative data from national Swedish routine registers held by Statistics Sweden, the Swedish National Council for Crime Prevention, and the Swedish National Board of Health and Welfare. In accordance with Swedish data protection laws, such administrative data are available only for specific research projects and cannot be shared with other researchers. Standard procedures for the release of Swedish administrative register data apply “https://www.scb.se/en/services/ordering-data-and-statistics/ordering-microdata/ (accessed on 27 March 2025)”, including permission from the Swedish Ethical Review Authority.

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
