# Peer review of "Parental Death and Premature Mortality in Individuals with Out-of-Home Care Experience in Sweden: A Nationwide Cohort Study"

_ijerph, 2025, doi:10.3390/ijerph22040580_

Round 1

Reviewer 1 Report

Comments and Suggestions for Authors

Thank you for this manuscript. In general, in my opinion, this is a clear and professionally prepared paper. I have just few suggestions how to improve it:

  1. I would recommend adding more keywords (up to 10) for better indexing.
  2. Please indicate a population in the title (i.e., Swedish) as A Nationwide Cohort Study is unspecific towards nationality.
  3. Line 165: Missing data: Please could you elaborate more on this? Excluding about 80000 participants because of missing data should be clarified. Please elaborate on the data quality.
  4. "Standard descriptive statistics of the study variables for men and women are reported separately...". Please elaborate whether non-binary people were analyzed. In such a large sample, it seems this identity could be presented.
  5. I would suggest avoiding very long paragraphs. Please increase the readability by dividing them.
  6. Please add references for all statistical analyses used in this study. Only several analyses were justified by the relevant literature.
  7. "Data Availability Statement: The Public Access to Information and Secrecy Act in Sweden prohibits the authors from making individual-level register data publicly available". Please elaborate whether the data are available upon request?
  8. Conclusions: I would suggest present more specific conclusions regarding the main findings of this study. The current conclusions focus on practical implication and future directions, and seem to be less related to the current findings (in contrast, the abstract of the paper is specific). Perhaps, this could be helpful: https://www.newcastle.edu.au/__data/assets/pdf_file/0009/333765/LD-Conclusions-LH.pdf 

Reviewer 2 Report

Comments and Suggestions for Authors

This paper is well-done with important evidence regarding the effects of out-of-home care.  There are some suggestions for improvement:

1)  The terms "premature mortality" and "all-cause mortality" should be defined when they are introduced in the Introduction.

2)  Lines 86-87 discuss a reduction in mothers' mental health, which triggers premature mortality.  Is this the child's or the mother's mortality?

3)  For lines 90-93, is there research to cite regarding the percentage of biological mothers who have substance abuse issues when their children are in out-of-home care?

4)  The fact that the hypotheses are opposites poses a problem.  If one hypothesis is disproven, the other hypothesis is supported.  There should only be one hypothesis that can either be supported or rejected.

5)  For Section 2.2, there is a large imbalance between the sample size for the non-OHC group and the OHC group.  This issue could be responsible for the lack of statistically significant findings and should be listed as a limitation.

6)  What were the sample sizes for each of the 5 categories of placements?

7) What was the sample size for children who experienced a parental death?

8)  Under confounding factors, it is confusing as to whether non-Swedish participants were included in the study, as the paper states it is a Swedish study.

9)  Could characteristics of the biological parent, such as education level, be confounders?

10) It seems that there may be a difference between the death of a parent who was the original custodial parent vs. a non-custodial parent.  Could this issue affect results?

Round 2

Reviewer 1 Report

Comments and Suggestions for Authors

Thank you for your professional replies and the work done during the revisions. I feel the paper is ready to be published.